# In situ single-crystal synchrotron X-ray diffraction studies of biologically active gases in metal-organic frameworks

Russell M. Main [1], Simon M. Vornholt [2], Cameron M. Rice [1], Caroline Elliott[1], Samantha E. Russell[1], Peter J. Kerr[1], Mark R. Warren[3] & Russell E. Morris [1 ✉]

Metal-organic frameworks (MOFs) are well known for their ability to adsorb various gases. The use of MOFs for the storage and release of biologically active gases, particularly nitric oxide (NO) and carbon monoxide (CO), has been a subject of interest. To elucidate the binding mechanisms and geometry of these gases, an in situ single crystal X-ray diffraction (scXRD) study using synchrotron radiation at Diamond Light Source has been performed on a set of MOFs that display promising gas adsorption properties. NO and CO, were introduced into activated Ni-CPO-27 and the related Co-4,6-dihydroxyisophthalate (Co-4,6-dhip). Both MOFs show strong binding affinity towards CO and NO, however CO suffers more from competitive co-adsorption of water. Additionally, we show that morphology can play an important role in the ease of dehydration for these two systems.

[1] EaStCHEM School of Chemistry, Purdie Building, North Haugh, St Andrews KY16 9ST, UK. [2] Department of Chemistry, SUNY Stony Brook, 100 Nicolls Road, 104 Chemistry, Stony Brook, NY 11790-3400, USA. [3] Diamond Light Source Ltd, Diamond House, Harwell Science & Innovation Campus, Didcot OX11 0DE, UK. ✉email: rem1@st-andrews.ac.uk

Metal-organic frameworks (MOFs) are an extensively researched and exciting group of materials, with a wide range of applications including the storage and release of biologically active gases[1]. Their structures consist of metal nodes or oxoclusters bound by organic linkers to create large open lattice frameworks[2]. MOFs are well known for their high porosities and exceptional gas adsorption properties, due to their tuneable node chemistry and exceptionally high specific surface areas[3,4].

CPO-27[5], also referred to as MOF-74[6], is a well-studied and widely used MOF comprising $M^{2+}$ metal ions, where M can be a wide range of metals such as Mg, Co, Ni, Cu and Zn, linked by 2,5-dihydroxyterephthalate (2,5-dhtp) linkers[7–9]. It crystallises in the R-3 space group and consists of hexagonal channels running along the crystallographic c-axis. These channels contain open metal sites upon removal of covalently bound water molecules, as such CPO-27 is an attractive option for the storage of gases, particularly Lewis acids such as CO and NO. A structural conformer of CPO-27, $M_2$(m-dobdc) where M is Co can be obtained by using the linker 4,6-dihydroxyisophthalic acid (4,6-dhip, an isomer of 2,5-dhtp)[10]. We refer to this material as Co-4,6-dhip in the rest of this paper. The isomeric linker commands a slightly altered short range structure in the metal node, which results in minutely distorted hexagonal channels and a different space group (R3m), but otherwise almost identical structural features (pore chemistry, and available open metal sites).

Gas storage and sequestration is amongst the most exciting and well researched areas of MOFs. The adsorption of relevant gases for energy production, e.g. $H_2$[11] and $CH_4$[12], the storage of pollutants such as $CO_2$[13], $SO_x$[14] and $NO_x$[15] have received much attention in recent years. However, the exact binding mechanisms of these gases often remain elusive. Another avenue is the storage and release of biologically active gases. There has been significant research into using MOFs to store and release NO[1,7,16–23]. NO is an important signalling molecule in mammalian physiology, serving a vital function in various systems throughout the body[24]. In high concentrations NO is toxic, but in low concentrations NO may be used in medical technologies[1,25,26]. MOFs such as CPO-27 have shown great potential in storing NO and the subsequent controlled release in the presence of water[1]. However, NO is not the only gas capable of performing this function and CPO-27 not the only MOF. Gases such as CO also have medical uses, and MOFs such as BioMIL-3[16], Co-4,6-dhip and a variety of zeolites have also been studied[27,28].

To fully understand the uptake and release thermodynamics it is important to know exactly how the gases bind within the frameworks. Single crystal X-ray diffraction (scXRD) provides invaluable understanding of the binding motif in these materials. By using the high intensity and high energy X-rays available at a synchrotron[29], it is possible to obtain, and model, data of high enough resolution to observe gas molecules within the MOF pore environment. We have previously published data on NO binding in Ni-CPO-27[30]. In this study we investigate the binding mechanism of CO and NO in Ni-CPO-27 and for the first time of its structural conformer MOF Co-4,6-dhip. An efficient activation protocol is determined and the MOF gas adsorption properties towards CO and NO are probed as previous work showed a reduced release of NO in the Co/Ni-4,6-dhip framework, while other works have disclosed a higher affinity of this framework toward $H_2$[10,31], when compared to CPO-27.

## Results

Each selected crystal was first activated at high temperature (450–500 K) and vacuum ($3.3 \times 10^{-6}$ mbar on the vacuum gauge). Samples were considered activated when the M—$O_{water}$

($O_w$) oxygen residual occupancy fell below 10%. However, complete activation, i.e., a residual occupancy of $O_w$ ~0% could not be achieved even after treatment under high vacuum and temperature; a similar behaviour was also observed in a previous study[30]. There are several possible reasons for this. Some residual water may be trapped in blocked one-dimensional channels on surface contact of the crystal with the glue used to hold it in place. It was also found that if the activated samples were cooled, the amount of bound water increased even under (dynamic) vacuum. This suggests that, because the whole gas delivery system cannot be heated, some water remains bound on the cool surfaces of the gas system even when the crystal itself is heated and this water is readsorbed once the crystal is cooled. Previous studies on other MOFs have successfully achieved complete dehydration[32], however the high number of open metal sites in these frameworks makes this more challenging to achieve. To mitigate this to some degree (but not completely), the crystal was exposed to the gases of interest while it was still at the dehydration temperature. However, a small amount of residual water will remain on gas uptake. The samples were then cooled to 300 K to observe how the gas binding changes and to gain better quality data. Presented here are the obtained structures. A list of experimental details with corresponding refinement quality factors (R1) can be found in supplementary table 1 and the full structure determination details have been deposited with the Cambridge Crystal Structure Database as described below.

Analysis on bulk samples of Ni-CPO-27 and Co-4,6-dhip can be found within the Supplementary Information. More detailed information on the two MOFs used in this study can be found in the publication by Vornholt et al. which also includes a number of different gas adsorption isotherms for the two systems[30].

## CO adsorption in Ni-CPO-27

*Activation.* In a first gas loading experiment, the uptake of CO in Ni-CPO-27, the crystal was first activated at 450 K for 5 h in vacuo. The activated structure is shown in Fig. 1 and supplementary data 6, where the framework shows the expected R-3 space group and characteristic hexagonal channels seen in Ni-CPO-27[30]. $O_w$ shows an occupancy of 6.06(2)%, the framework is therefore considered activated.

*Gas loading.* CO was introduced to the crystal at 450 K at an absolute pressure of 2.5 bar. Figure 1, and Supplementary Data 3, shows that CO has chemisorbed onto the vacant metal site, via the C atom with a linear geometry and 38(3)% partial occupancy. The CO bond length is 1.17(7) Å, within error of the 1.13 Å bond length computed for molecular CO[33]. The Ni—C bond length is 2.16(3) Å, longer than has been recorded spectroscopically on Ni surfaces[34], but of the same order as that found for Co-CPO-27 at lower temperature[19]. The Ni-C-O angle is 176.388° approximately linear binding. As shown and discussed later in more detail (Fig. 2), the CO appears to be disordered. Enlarged atomic displacement parameters (ADPs) will be a result of the atomic displacement at the high temperature at which the dataset was recorded. Additionally, despite the structure being deemed activated any disorder between the CO and residual water bound to the metal site may artificially elongate the modelled Ni-C bond and enlarge obtained ADPs for the guest. However, a sensible refinement of multiple oxygen position to resolve any such disorder was not successful for this dataset.

Reducing the temperature to 300 K with the same CO pressure has two effects visualized in Fig. 1 and seen in supplementary data 7. Firstly, the CO loading increases to 63(4)%. With this, the CO bond at 1.16(4) Å is within the error of that from the 450 K structure, the Ni—C bond is also similar at 2.12(4) Å and the Ni-

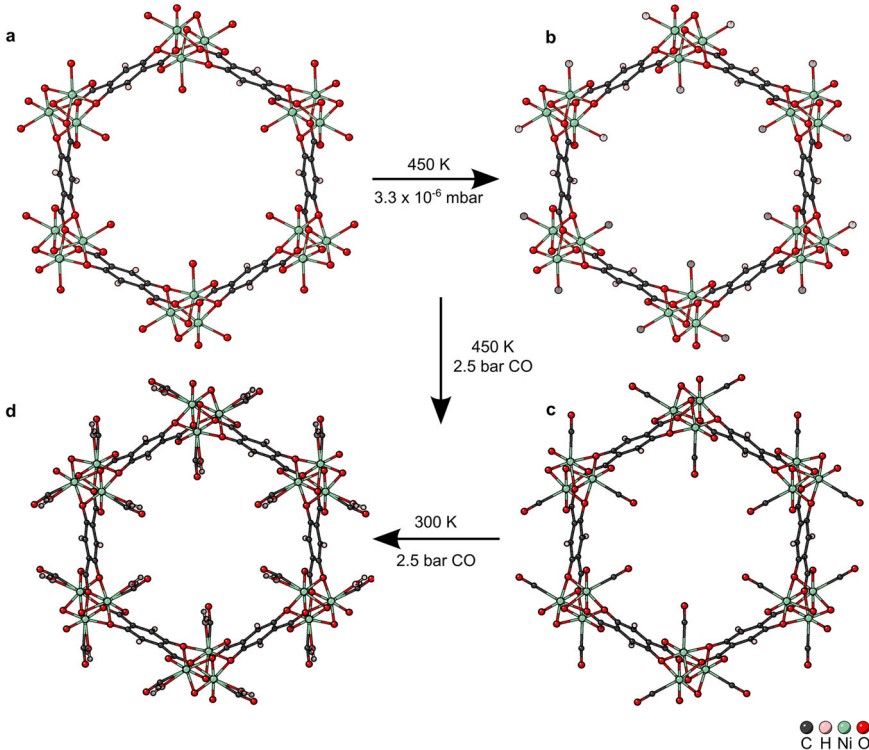

**Fig. 1 Depictions of the crystal structure of Ni-CPO-27 along the crystallographic *c*-axis under different gas loading conditions.** The framework is shown with 40% space filling spheres. **a** Hydrated structure at room temperature and ambient pressure. **b** Activated structure at 450 K and $3.3 \times 10^{-6}$ mbar, with 6.06(2)% water occupancy, which is considered activated, CCDC number: 2208832. **c** CO loaded structure at 450 K and 2.5 bar CO pressure, with 38(3)% CO occupancy, CCDC number: 2208829. **d** CO loaded structure at 300 K and 2.5 bar CO pressure, with 63(4)% CO occupancy, and 27(3)% water occupancy, CCDC number: 2208833.

C-O bond angle changes to 168.6°. Secondly, some competing water also binds to the open metal site with 27(3)% loading and a Ni—$O_w$ bond length of 2.16(7) Å. A similar effect was seen with NO in Ni-CPO-27[30]. This is not unexcepted nor undesirable as the competition between water and NO is what allows controlled release[17]. A reduction in CO bond length on cooling was expected, as Long and co-workers found spectroscopically that CO in Ni-CPO-27 was adsorbed non-classically and was blue shifted on binding[35]. However, the presence of water in the model make such variations hard to uncover unambiguously. In addition, the modelled water is likely the main cause of the deviation in Ni-CO bond angle, as modelling the water and CO from the same region of electronic density causes the C to be slightly displaced. Therefore, it is probable that the CO still prefers the linear geometry seen at higher temperatures. The CO molecule was less disordered at 300 K than at higher temperature and yielded a model with only one site (Fig. 2).

*Physisorbed CO in Ni-CPO-27.* It was not possible to sensibly refine any physisorbed gas molecules. However, by calculating a mask with a 1.2 Å probe it was possible to estimate the open pore volume and the amount of $e^-$ density remaining within the pores, a useful proxy for the amount of gas molecules present in the pore. The activated sample had a free pore volume of 2097 Å³ per unit cell. Upon loading with CO this reduced to 1782 Å³ per unit cell, and cooling reduced it slightly to 1770 Å³ per unit cell, this is approximately 75% of the total, theoretical, dehydrated pore size. The $e^-$ density in the pore of the activated sample was estimated at 0.147 $e^-$ per Å³. Loading with CO increased this to 0.535 $e^-$ per Å³ and cooling slightly reduced it to 0.531 $e^-$ per Å³. This is the expected pattern as introducing CO into the pores will increase the $e^-$ density, cooling increases the amount of CO chemisorbed

so there is now less free in the pore. From these values it can be estimated that there are roughly 48 CO molecules per unit cell tumbling freely within the pore at 2.5 bar pressure and 300 K.

## CO adsorption in Co-4,6-dhip

*Activation.* Following the CO loading of Ni-CPO-27, CO loading of Co-4,6-dhip was performed to identify any differences between the two systems. Dehydration of a Co-4,6-dhip crystal was more complex than the Ni-CPO-27 sample, involving prolonged treatment at 450 K followed by a slow ramp over 10 h up to 500 K and holding at temperature for 1.5 h in vacuo (Fig. S3). This was followed by flushing the sample with CO at 2.5 bar and then removing it in vacuo (Fig. S3). Figure 3 and Supplementary Data 2 shows the resulting activated Co-4,6-dhip with the expected *R3m* symmetry[10]. The $O_w$ occupancy on the Co was 9.0(4)%, higher than that achieved for the dehydration of Ni-CPO-27. The significant difference in dehydration conditions required is likely due to the differences in morphology, as one might expect Ni to be harder to dehydrate due to its higher hydration enthalpy[36], as well as the very similar thermal gravimetric analysis profiles of the two MOFs between 300 and 470 K (Fig. S4). Co-4,6-dhip crystallizes in long needles, whereas Ni-CPO-27 forms broader hexagonal rods (Fig. S5)[30]. If the hexagonal channels run down the length of the Co-4,6-dhip needle then the available surface area from which to lose water will be significantly reduced, therefore making dehydration harder.

*Gas loading.* CO was introduced to the crystal at 500 K and 2.5 bar. This caused CO to be chemisorbed to the open cobalt site, via the C atom; however, it also introduced water, as seen in Fig. 3 and Supplementary Data 1. This caused the CO loading to be

only 10.7(15)% with 15.2(15)% water loading. The CO bond length is 1.13(3) Å, the Co—C bond length is 2.26(4) Å and the Co-C-O bond angle is 151.747°. The Co—$O_w$ bond length is 2.22(2) Å. The increase in Co—CO bond length and deviation in bond angle may be due to a weaker bond but the higher temperature and large proportion of water will affect these values significantly and further spectroscopy is needed to verify any changes. Again, the CO is highly disordered, and it was not

possible to model sensibly as highlighted in Fig. 2. This binding behavior corresponds with results obtained by Long and co-workers who found similar bond lengths and geometry at lower temperatures in Co-CPO-27[37]. Upon cooling the sample, a black residue formed on the outer wall of the gas cell and no further data could be gathered.

*Physisorbed CO in Co-4,6-dhip.* It was not possible to sensibly refine any physisorbed CO molecules. However, by calculating a mask with a 1.2 Å probe it was possible to estimate the open pore volume and the amount of $e^-$ density within the pores. The activated sample had a pore volume of 2061 Å³ per unit cell and $e^-$ density of 0.568 $e^-$ per Å³, higher than expected under the conditions. On loading the sample, the pore volume changed to 1830 Å³ per unit cell and the $e^-$ density to 0.644 $e^-$ per Å³. The free pore volume calculated here is approximately 78% of the total, theoretical, dehydrated pore volume. The high $e^-$ densities may be explained by disorder within the crystal structure, or even framework decomposition, placing apparent $e^-$ density within the pores. Due to the high $e^-$ density of the activated sample an accurate measure of the number of physisorbed gas molecules is not possible.

### NO adsorption in Co-4,6-dhip

*Activation.* Lastly, NO loading of the Co-4,6-dhip system was studied, allowing a direct comparison of the binding of different gases in this system and any changes of the NO binding in Co-4,6-dhip compared to Ni-CPO-27[30]. This crystal activated more easily than the previous, with a max temperature of only 450 K in vacuo required with no need for gas flushing. A possible explanation for this is the position of the glue relative to the crystal. The previous sample had glue at one end of the needle, whereas this sample had the glue in the middle. Since the hexagonal channels run down the length of the needle it would explain the difference in dehydration conditions needed for the same structure as more pores are blocked when the crystal is mounted end on. It is also possible that this difference is due to slight structural variations between the crystals. The residual occupancy of $O_w$ was 7.7(4)% (Fig. 4 and Supplementary Data 5), and the crystal structure was as expected and discussed above.

*Gas loading.* Loading NO at 450 K and 2.5 bar, caused NO to become chemisorbed to the open metal site, via the N atom, with 84.5(15)% loading (Fig. 4 and Supplementary Data 8). As was also

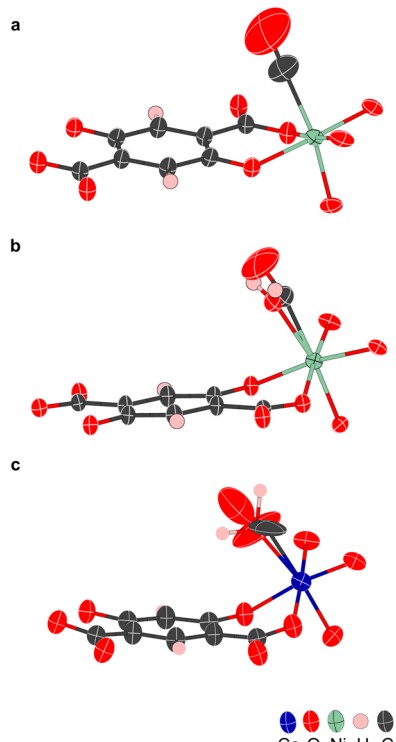

**Fig. 2 The asymmetric units of Ni-CPO-27 and Co-4,6-dhip, showing 50% probability ellipsoids, highlighting the binding of CO in the two systems. a** Ni-CPO-27 at 450 K with 2.5 bar CO pressure, with 40(4)% CO occupancy. **b** Ni-CPO-27 at 300 K with 2.5 bar CO pressure, with 63(4)% CO occupancy and 27(3)% water. **c** Co-4,6-dhip at 500 K with 2.5 bar CO pressure, with 10.7(15)% CO occupancy and 15.2(15)% water occupancy.

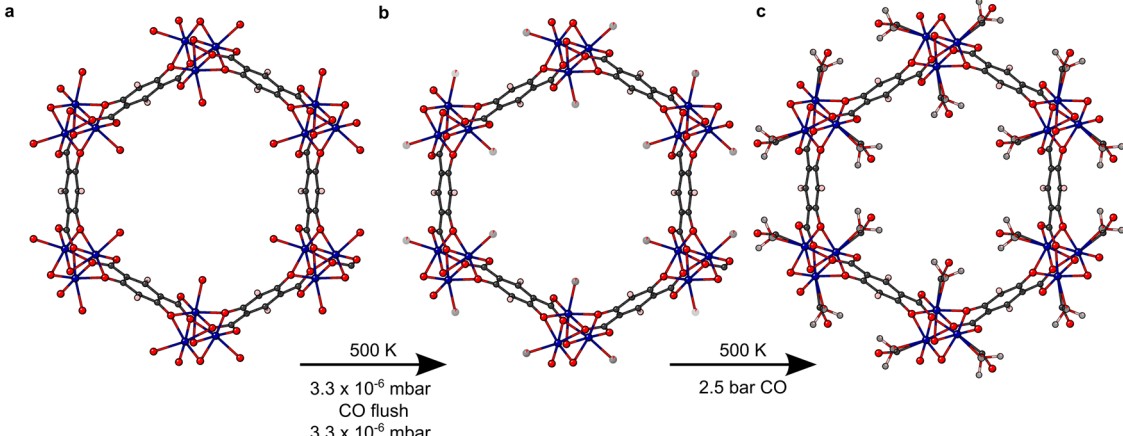

**Fig. 3 Depictions of the crystal structure of Co-4,6-dhip along the crystallographic *c*-axis under different gas loading conditions, with 40% space filling spheres. a** Hydrated structure at room temperature and ambient pressure. **b** activated structure at 500 K and $3.3 \times 10^{-6}$ mbar, with 9.0(4)% water occupancy, CCDC number: 2208828. **c** CO loaded structure at 500 K and 2.5 bar CO pressure, with 10.7(15)% CO loading and 15.2(15)% water, CCDC number: 2208827.

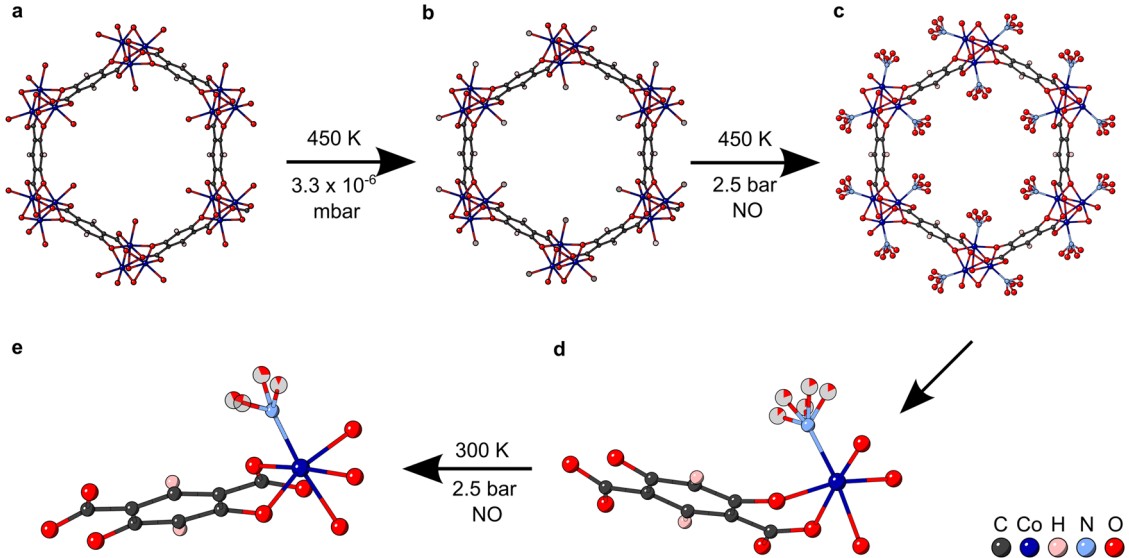

**Fig. 4 Depiction of the crystal structure of Co-4,6-dhip, under different gas loading conditions, along the crystallographic *c*-axis with 40% space filling spheres and as asymmetric units with 50% probability ellipsoids. a** Hydrated structure at room temperature and ambient pressure. **b** activated structure at 450 K and 3.3 × 10⁻⁶ mbar, with 7.7(4)% water occupancy, CCDC number: 2208831. **c** NO loaded structure at 450 K and 2.5 bar NO pressure, with 84.5(15)% NO loading, CCDC number: 2208834. The asymmetric unit space filling spheres set at the 40% level is shown in: **d** NO loaded structure at 450 K and 2.5 bar NO pressure, with 84.5(15)% NO loading. **e** NO loaded structure at 300 K and 2.5 bar NO pressure, with 92(3)% NO loading, CCDC number: 2208830.

observed with Ni-CPO-27[30], the NO was disordered and was modelled with five O environments around the N atom (Fig. 4d). To maintain a sensible geometry the refinement included restraints on the N-O bonds as additional observations which resulted in N-O distances that ranged from 1.12(2)–1.15(2) Å. The unrestrained Co—N bond length was 1.972(10) Å. The NO binds in a bent geometry; again this was also seen with Ni-CPO-27[30].

Cooling the sample to 300 K under the same NO pressure caused an increase in NO loading to 92(3)% (Fig. 4 and Supplementary Data 4). The O atom was again disordered and was modelled with four O environments around the N. This reduction in disorder may be down to the reduction in temperature, as seen previously[30]. Unfortunately, this data set is weaker due to increased background noise, likely caused competition at lower temperature with diffraction from the gas cell capillary (Fig. 4e). The N-O interatomic distance was restrained in the same manner as above and resulted in refined distances that ranged from 1.11(2)–1.14(3) Å. The Co—N bond length was 1.960(11) Å, within error of the higher temperature sample. The NO still prefers the bent geometry.

*Physisorbed NO in Co-4,6-dhip.* Again, modelling physisorbed NO was not possible but the mask calculated with a 1.2 Å probe estimated the following parameters. The activated sample had a pore volume of 2061 Å³ per unit cell which decreased to 1434 Å³ per unit cell when NO was added and decreased further to 1425 Å³ per unit cell on cooling. This is approximately 60% of the total, theoretical, dehydrated pore volume. The e⁻ density in the pore was again surprisingly high for the activated sample at 0.578 e⁻ per Å³, it increased to 0.785 e⁻ per Å³ when NO was added and dropped to 0.648 e⁻ per Å³ on cooling. See above for explanation of high e⁻ densities.

## Conclusion

We have verified that Ni-CPO-27 can be activated successfully at 450 K under high vacuum. Co-4,6-dhip can be more difficult to

activate, possibly due to its needle like morphology, restricting the rate of water loss from the internal channels. For the first time we show that NO can bind in Co-4,6-dhip successfully and shows many similar characteristics to how it binds within Ni-CPO-27[30]. CO can also bind at the open metal site of both Ni-CPO-27 and Co-4,6-dhip. The CO bond remains similar to molecular CO suggesting that the metal binding involves very little redistribution of e⁻ density. CO also prefers binding in a linear geometry compared to NO which prefers a bent geometry. These bonding characteristics match well with the traditional orbital bonding model of CO and NO[38]. This work further elucidates the importance of high quality, in situ, scXRD experiments to understand not only how gases bind within molecules; but how subtle changes to linker and metal nodes can affect this binding. Having high quality CIF files as definitive structure solution of guest-loaded MOFs is an invaluable for further analysis such as computational modelling that relies on exact bond distances. It is important to note how the relative competition between an adsorbed gas molecule and water can impact on potential utility of a material, especially for biological gases such as NO and CO. The more significant competition between CO and H₂O that has been revealed by this work will have an effect on how a CO-loaded MOF will behave under physiological (aqueous) conditions. This in itself indicates the importance of undertaking complex and advanced characterisation experiments to understand these interactions more thoroughly.

## Methods

**Synthesis**. Single crystals of Ni-CPO-27 and Co-4,6-dhip were synthesised by the literature procedure produced by Vornholt et al.[30].

For Ni-CPO-27, Nickel acetate tetrahydrate (1 mmol) was dissolved in water (30 mL) and added to a Teflon liner (50 mL). 2,5-dihydroxyterephthalic acid (0.5 mmol) and 4,6-dihydroxyterephthalic acid (0.5 mmol) were added to the liner and left to stir for 15 min. The liner was then capped, sealed in an autoclave and placed in the oven for 3 days at 130 °C. Yellow-brown, rectangular rods of CPO-27-Ni, with a yield of 73% (based on nickel salt), were obtained after filtration.

For Co-4,6-dhip, in a Teflon liner (30 mL), cobalt acetate tetrahydrate (1 mmol) and benzoic acid (4 mmol) were dissolved in a water/ethanol solvent mixture (5 mL each). The linker 4,6-dhip (1 mmol) was dissolved in THF (10 mL), slowly added to the salt/modulator mixture, and left to stir for 15 min. The Teflon liner was then

capped and left to react for 3 days at 150 °C. Pink single crystals were obtained after filtration (70% yield based on Co).

Phase purity was verified by powder X-ray diffraction, Figs. S1 and S2, and morphology by scanning electron microscopy, Fig. S5.

**In-situ single crystal diffraction studies**. In-situ gas cell diffraction experiment on single crystals were carried out on the four-circle Newport diffractometer equipped with a Dectris Pilatus 300 K detector in I19-2 beamline, Diamond Light Source. A wavelength of 0.48590 Å (Ag K-edge) was utilized to give a complete dataset from a single 340 degree phi sweep (1700 images, 0.2 deg/image). Selected crystals were mounted with a mitogen mount (50 μm) and were secured with a non-diffracting two component epoxy glue (LOCKTITE DOUBLE BUBBLE™). Care was taken to use as little glue as possible to avoid blocking any channels and ensure good gas transport through the crystal. For gas cell experiments, the crystal mount was inserted into a pre-assembled gas cell, with super glue used to hold the mount securely in place in the gas cell capillary. The gas cell was then sealed using the Swagelok mechanism and leak tested. The activation temperature ranged from 450–500 K, with a heating ramp of 360 K/h, in vacuo $(3.1 \times 10^{-6}$ mbar at the pump). A data collection at 300 K of the activated systems was obtained for comparison purposes. Activated crystals were exposed to 2.5 bar of the selected gas (CO or NO) at 450–500 K and data collected after 30 min exposure. The sample was then cooled to 300 K at the same gas pressure and another data set collected. After the crystal had been exposed to the gas for another 30 min, an additional dataset was obtained to monitor gas uptake.

**Data processing**. Data collection were setup using the general data acquisition (GDA) software and were automatically processed using xia2[39] with DIALS[40] routines. Subsequently, Olex2 GUI[41] (with shelXT[42] as solution and shelXL[43] as refinement tool) was used for structure solution and refinement, respectively. Obtained crystal structures were visualised using the CrystalMaker software kit[44]. Special refinement details can be found in the Supplementary Methods.

## Data availability

The research data supporting this publication can be accessed at https://doi.org/10.17630/fbdce016-623f-48f3-898c-f95216eaaab1 or from the authors on request. The X-ray crystallographic coordinates for structures reported in this study have been deposited at the Cambridge Crystallographic Data Centre (CCDC), under deposition numbers 2208827 – 2208834. These data can be obtained free of charge from The Cambridge Crystallographic Data Centre via www.ccdc.cam.ac.uk/data_request/cif. CIF files are available as supplementary data files 1–8.

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

## Acknowledgements
The authors gratefully acknowledge the assistance of Diamond Light Source, particularly the staff of beamline I-19. The data was obtained from visit cy29217-1. The authors are also grateful for financial assistance from the ERC under advanced grant 787073, the EPSRC for a studentship (EP/N509759/1) and the CRITICAT Centre for Doctoral Training (EP/L016419/1). In addition, we gratefully acknowledge the help of Matthew Kapelewski and the Jeff Long group for help with the synthesis of 4,6-dhip.

## Author contributions
The manuscript was written through contributions of all authors. All authors have given approval to the final version of the manuscript. R.M.M. Contributed to the in situ experiment, data analysis and preparation of first draft. S.M.V. Contributed to the proposal, synthesis and preparation of first draft. C.M.R. contributed to the in-situ experiment. C.E. contributed to the in situ experiment. S.E.R. contributed to the in situ experiment and preparation of first draft. P.J.K. contributed to the synthesis. M.R.W contributed to the in situ experiment, data analysis and preparation of first draft. R.E.M. contributed to the proposal and obtaining the listed funding.

## Competing interests
The authors declare no competing interests.
