## [Peer Review File · Communications Chemistry]

Reviewers' comments:

Reviewer #1 (Remarks to the Author):

In this manuscript entitled "In Situ single-crystal synchrotron X-ray Diffraction Studies of Biologically Active Gases in Metal-Organic Frameworks", authors have performed an in-Situ single-crystal synchrotron X-ray Diffraction of nitric oxide (NO) and carbon monoxide (CO) gases in MOF-74 (Ni and Co ions)

1) Although authors have used the term dehydrated for the structure MOF-74 (Ni and Co ions) collected at 450 K under vacuum but at the same time, they have mentioned "if the dehydrated samples were cooled, the amount of bound water increased even under (dynamic) vacuum." In the crystal structure of dehydrated form, it shows a water occupancy of 5.9(19)%. Considering the above statement and the amount of water present in the dehydrated framework, the term dehydrated is something the author needs to reconsider. The term partially hydrated form is better suited instead of dehydrated form as the framework still contains some water molecules even after activation under vacuum. In this regard, recently, Lama et. al has published a very nice example of activating a MOF to get the dehydrated form using a conventional single crystal X-ray diffraction study (Inorg. Chem. 2022, 61, 939-943).

2) MOF-74 is a porous material. Under gas pressure, the authors have modeled and mentioned only about the chemically absorbed gas molecule. What about the physically absorbed molecules inside the pore of the framework? Did the author try to model those physically absorbed gas molecules as well?

3) Even though the authors have mentioned about the electron density calculation using a probe radius of 1.2 Å but in all the CIF files, no such squeeze analysis data is appended. These data should be included in the revised CIF files to have a comparative idea of the number of electrons present in the pores of an activated form and the gas pressurize structures.

4) Getting an activated form and modeling gas molecules using single-crystal diffraction is always difficult. From the single crystal structure data, it was observed that the reliability factor (R1) for an activated form of Ni-MOF74 is almost three times higher as compared to the activated structure of Co-MOF. Is there a suitable justification regarding this difference in R1 value?

Overall, the work is very interesting and always challenging. I recommend publishing this work in this journal after considering the above points in the revised manuscript.

Reviewer #2 (Remarks to the Author):

The work describes a comprehensive structural study of two MOFs that incorporate open metal sites as a function of loading of CO or NO. This was achieved by in situ synchrotron single crystal X-ray diffraction. Although there are extensive studies on gas binding in MOFs over the recent past, detailed structural characterisation of binding of NO/CO remains to be rare. Studies on these gases have impacts on both clean air and biological applications.

Full details on the conduction of the experiments and refinements of the data have been presented. I particularly welcome the transparent discussions given by the authors, for example, on the analysis of the retained water molecules, presence of structural disorder of guest molecules, competitive adsorption. From my own experience, such experiments are highly challenging. The manuscript is quite knowledgeable to readers who are new to this field. The binding of NO/CO in Ni/Co-CPO-27 has been previously studied by powder diffraction and the related material Co-4,6-dhp was investigated here for the first time. Because of the advantage of single crystal diffraction, a great deal of in-depth

structural information has been obtained, esp. for the analysis of competitive adsorption and subtle changes on the local framework. Overall, the work illustrates a nice example of the advanced structural study in porous MOFs and I support the publication.

Reviewer #3 (Remarks to the Author):

The article, "In Situ single-crystal synchrotron X-ray Diffraction Studies of Biologically Active Gases in Metal-Organic Frameworks", reports on the in situ structure determinations of two nearly fully dehydrated MOFs, as well as their structures when they have been exposed to CO and NO gases. Although the crystallographic work seems to be handled competently (barring a few minor points raised below), these studies by themselves do not constitute sufficient work for publication, in my opinion. For example, the authors make no reference at all to gas sorption studies that may have been carried out on bulk samples, in order to corroborate their findings of CO and NO binding in single crystals. Thermogravimetric analysis studies on bulk samples, or reference to these, may also have been useful in determining expected conditions under which the crystals would be fully dehydrated. I have a few comments below, should the authors find them useful for a future submission.

Comments:

1. Pg. 4, line 81 – I am not sure that the glue may be the reason for the MOFs not being fully dehydrated. Even, if the glue blocks the channel from one side, the channels should still be able to be evacuated from the other end.
2. Pg 5, line 96 – The name of the space group should be given, not just the number.
3. Atoms belonging to the same molecule, should be modelled with the same site occupancy factors. For example, the carbon and oxygen atom in structure CCDC 2208829 have different occupancies. This has been correctly done elsewhere, however, I would also suggest that for the disordered NO molecules that the total occupancies of the oxygen atoms be constrained to add up to the occupancy of the nitrogen atom.
4. The higher water content of the dehydrated Co-4,6-dhip structure, as compared to that of the Ni-CPO-27 structure, is attributed to the needle-shaped crystals of the Co MOF which have smaller surface areas for dehydration. Should this be the reason, then the crystal could have been dehydrated for longer. However, the difference in hydration (7% vs 9%) is marginal, it simply could also just be that the chosen Co crystal gave that value and that another may have a slightly different degree of dehydration.
5. The authors did not discuss the CO/NO molecules that may be trapped in the channels (at the various pressures and temperatures) but not adsorbed onto a metal site. Perhaps, the reference to the residual electron density in the unit cell was a reference to this but it was not explicitly discussed. These molecules (present in the channel but not adsorbed) may not be deemed relevant in terms of the motivation of the study but then this should be stated.
6. Quoting electron densities such as "0.147 e- Å³" is rather meaningless to the reader. There should be an attempt to relate these values to the chemical entities that are being adsorbed.
7. Pg. 11, line 183 – I do not find the reason given for high electron densities, i.e., interference of the glue, a plausible reason.

8. The supplementary information states that the oxygen atoms of water molecules in dehydrated structures were refined isotropically (although in CCDC 2208832 the water molecule oxygen atom was refined anisotropically). Why is this the case when the disordered water molecules were refined anisotropically? Also why are the hydrogen atoms not added for the water molecules in the dehydrated crystals, but were added when water was modelled as being disordered with CO?

9. I think it is informative to also cite the percentage void spaces, together with the absolute void volume per unit cell.

10. If the point of contact with the glue is crucial and mounting the crystal on its 'side' (i.e., in the middle) is more beneficial to dehydration, then the authors should have repeated the other experiments with the crystal mounted in this way. However, as stated earlier, I am not convinced that this is the reason and that perhaps there could be a variation from crystal to crystal.

In the following our changes and responses are in red italics.

Reviewer #1 (Remarks to the Author):

In this manuscript entitled "In Situ single-crystal synchrotron X-ray Diffraction Studies of Biologically Active Gases in Metal-Organic Frameworks", authors have performed an in-Situ single-crystal synchrotron X-ray Diffraction of nitric oxide (NO) and carbon monoxide (CO) gases in MOF-74 (Ni and Co ions)

1) Although authors have used the term dehydrated for the structure MOF-74 (Ni and Co ions) collected at 450 K under vacuum but at the same time, they have mentioned "if the dehydrated samples were cooled, the amount of bound water increased even under (dynamic) vacuum." In the crystal structure of dehydrated form, it shows a water occupancy of 5.9(19)%. Considering the above statement and the amount of water present in the dehydrated framework, the term dehydrated is something the author needs to reconsider. The term partially hydrated form is better suited instead of dehydrated form as the framework still contains some water molecules even after activation under vacuum. In this regard, recently, Lama et. al has published a very nice example of activating a MOF to get the dehydrated form using a conventional single crystal X-ray diffraction study (Inorg. Chem. 2022, 61, 939-943).

We have changed the term dehydrated to activated where appropriate. We have added a sentence to section 2.0 noting that some MOFs can be completely dehydrated and added the reference requested. However, it should be understood that there is a significant difference between how strongly water is held on an open metal site (this work) compared to that published in the Inorg. Chem. Paper where there are no open metal sites.

2) MOF-74 is a porous material. Under gas pressure, the authors have modeled and mentioned only about the chemically absorbed gas molecule. What about the physically absorbed molecules inside the pore of the framework? Did the author try to model those physically absorbed gas molecules as well?

We have added to each relevant part of the result section, that attempts were made to model the physisorbed gases without success. We have expanded this section as well to take into account comments from reviewer 3, see below.

3) Even though the authors have mentioned about the electron density calculation using a probe radius of 1.2 Å but in all the CIF files, no such squeeze analysis data is appended. These data should be included in the revised CIF files to have a comparative idea of the

number of electrons present in the pores of an activated form and the gas pressurize structures.

The relevant mask data has been appended to CIF files as text within _mask_special_details as the mask was not used in the published structures (as is the norm)

4) Getting an activated form and modeling gas molecules using single-crystal diffraction is always difficult. From the single crystal structure data, it was observed that the reliability factor (R1) for an activated form of Ni-MOF74 is almost three times higher as compared to the activated structure of Co-MOF. Is there a suitable justification regarding this difference in R1 value?

The Co MOF was more strongly diffracting than the Ni MOF, and this better data set is likely behind the improved R1 value. We have no really strong argument apart from this simple crystal quality as to why this is so have not included this in the paper. Indeed it is well known that Ni MOF74 is much harder to make in single crystal form than the cobalt version.

Overall, the work is very interesting and always challenging. I recommend publishing this work in this journal after considering the above points in the revised manuscript.

Reviewer #3 (Remarks to the Author):

The article, "In Situ single-crystal synchrotron X-ray Diffraction Studies of Biologically Active Gases in Metal-Organic Frameworks", reports on the in situ structure determinations of two nearly fully dehydrated MOFs, as well as their structures when they have been exposed to CO and NO gases. Although the crystallographic work seems to be handled competently (barring a few minor points raised below), these studies by themselves do not constitute sufficient work for publication, in my opinion. For example, the authors make no reference at all to gas sorption studies that may have been carried out on bulk samples, in order to corroborate their findings of CO and NO binding in single crystals. Thermogravimetric analysis studies on bulk samples, or reference to these, may also have been useful in determining expected conditions under which the crystals would be fully dehydrated. I have a few comments below, should the authors find them useful for a future submission.

We have included TGA data on bulk samples within the supplementary information, and briefly discussed it while talking about dehydration differences. We have included a sentence in section 2.0 outlining that further analysis including isotherms can be found in previous papers.

Comments:

1. Pg. 4, line 81 – I am not sure that the glue may be the reason for the MOFs not being fully dehydrated. Even, if the glue blocks the channel from one side, the channels should still be able to be evacuated from the other end.

The reason for incomplete dehydration is due to the large amount of open metal sites as mentioned in section 2.0. The glue may affect the rate of dehydration as outlined by the increased temperature and time needed to activate this sample. Hence our suggestion of morphology and glue position effecting the kinetics of dehydration. However, we were, of course, speculating here.

2. Pg 5, line 96 – The name of the space group should be given, not just the number.

This has been corrected

3. Atoms belonging to the same molecule, should be modelled with the same site occupancy factors. For example, the carbon and oxygen atom in structure CCDC 2208829 have different occupancies. This has been correctly done elsewhere, however, I would also suggest that for the disordered NO molecules that the total occupancies of the oxygen atoms be constrained to add up to the occupancy of the nitrogen atom.

The Cif 2208829 has been corrected so that the C and O have the same occupancy. For the NO samples this has already been done, so that the sum of the O occupancies is within error of the N occupancy.

4. The higher water content of the dehydrated Co-4,6-dhip structure, as compared to that of the Ni-CPO-27 structure, is attributed to the needle-shaped crystals of the Co MOF which have smaller surface areas for dehydration. Should this be the reason, then the crystal could have been dehydrated for longer. However, the difference in hydration (7% vs 9%) is marginal, it simply could also just be that the chosen Co crystal gave that value and that another may have a slightly different degree of dehydration.

It is not that the Co-4,6-dhip has higher water content in the published structure but that it took a longer time and higher temperature to reach this value. This is why we conjectured that morphology was impacting the ease of dehydration. However, the referee makes a good point that the difference is small and probably not significant.

5. The authors did not discuss the CO/NO molecules that may be trapped in the channels (at the various pressures and temperatures) but not adsorbed onto a metal site. Perhaps, the reference to the residual electron density in the unit cell was a reference to this but it was not explicitly discussed. These molecules (present in the channel but not adsorbed) may not be deemed relevant in terms of the motivation of the study but then this should be stated.

We have added some additional information to the relevant results sections outlining that we couldn't model physisorbed gas molecules, as well as an additional explanation on how the mask calculations can be used as a proxy on how much physisorbed gas molecules are present.

6. Quoting electron densities such as "0.147 e- Å³" is rather meaningless to the reader. There should be an attempt to relate these values to the chemical entities that are being adsorbed.

Where possible we have included a rough estimate of how these values correspond to free gas molecules per unit cell.

7. Pg. 11, line 183 – I do not find the reason given for high electron densities, i.e., interference of the glue, a plausible reason.

We have removed any text suggesting that the glue has increased the electron densities.

8. The supplementary information states that the oxygen atoms of water molecules in dehydrated structures were refined isotropically (although in CCDC 2208832 the water molecule oxygen atom was refined anisotropically). Why is this the case when the disordered water molecules were refined anisotropically? Also why are the hydrogen atoms not added for the water molecules in the dehydrated crystals, but were added when water was modelled as being disordered with CO?

2208832 has been corrected to align with other samples. The reason for the simplistic water modelling is to allow easy comparison between multiple samples and reducing any electron density within the pores artificially inflating the modelled water content.

9. I think it is informative to also cite the percentage void spaces, together with the absolute void volume per unit cell.

Percentage void space has been added to the relevant sections.

10. If the point of contact with the glue is crucial and mounting the crystal on its 'side' (i.e., in the middle) is more beneficial to dehydration, then the authors should have repeated the other experiments with the crystal mounted in this way. However, as stated earlier, I am not convinced that this is the reason and that perhaps there could be a variation from crystal to crystal.

Due to time limitations at the synchrotron repeat experiments were not possible (and would be extremely difficult to justify on cost grounds). We have added a sentence saying that it is also possible that variation from crystal to crystal may explain the difference in dehydration conditions, which is undoubtedly true.

REVIEWERS' COMMENTS:

Reviewer #1 (Remarks to the Author):

The authors have modified the manuscript where needed and sufficient justifications/explanations are provided on all the comments given by the reviewers. I am happy to accept the manuscript in this present form.

Reviewer #3 (Remarks to the Author):

The article, "In Situ single-crystal synchrotron X-ray Diffraction Studies of Biologically Active Gases in Metal-Organic Frameworks", reports on the in situ structure determinations of two nearly fully dehydrated MOFs, as well as their structures when they have been exposed to CO and NO gases. I have reviewed the responses (to both reviewers) and changes that the authors have made. I appreciate that some of the issues raised are difficult to have exact answers for, but I'm satisfied with the explanations offered by the authors. The work is interesting and I am satisfied that the authors have addressed all the reviewers' comments. Thus, I support publication. However, I do have a few additional comments that arose due to the added information on the pore channel content.

Comment:

1. The authors stated:

"It was not possible to sensibly refine any physisorbed gas molecules. However, by calculating a mask with a 1.2 Å probe it was possible to estimate the open pore volume and the e- density remaining within the pores, a useful proxy for the amount of disordered gas present in the pore. The activated sample had a free pore volume of 2097 Å³/ unit cell. Upon loading with CO this reduced to 1782 Å³ / unit cell, and cooling reduced it slightly to 1770 Å³ / unit cell; this is approximately 50% of the total, theoretical, dehydrated pore size."

Comment: This presumably refers to when the ligated water molecule is deleted. According to MERCURY (for which values will be slightly different) the pore volume is then 2353 Å³ / unit cell, thus the pore volume of CO loaded at 300 K is much more than 50% of the theoretical dehydrated pore size.

2. The authors stated:

"The e- density in the pore of the activated sample was estimated at 0.147 e-/ Å³. Loading with CO increased this to 0.535 e-/ Å³ and cooling slightly reduced it to 0.531 e-/ Å³. This is the expected pattern as introducing CO into the pores will increase the e- density, cooling increases the amount of CO chemisorbed so there is now less free in the pore. From these values it can be estimated that there are roughly 24 CO molecules per unit cell tumbling freely within the pore at 2 bar pressure and 300 K."

Comment: I need some assistance in understanding the numbers. For CO loading at 300 K, the electron density was estimated to be 0.531 e-/ Å³. I presume this is multiplied by the quoted pore volume of 1770 Å³ in order to arrive at the total number of electrons? In other words, not the multiplied by the unit cell volume? If the former holds that will yield ~940 electrons for the pore per unit cell. CO has 14 electrons, how does this number correspond to 24 CO molecules per unit cell, because 14 x 24 = 336? The disparity is even worse if the residual electron density per Å³ is based on

the unit cell volume and not the pore volume. In this regard, I suggest that the authors add a section in the supplementary information to show how they have arrived at the estimation of the number of physisorbed gas molecules.

Response to reviewer #3

1. The authors stated:

“It was not possible to sensibly refine any physisorbed gas molecules. However, by calculating a mask with a 1.2 Å probe it was possible to estimate the open pore volume and the e- density remaining within the pores, a useful proxy for the amount of disordered gas present in the pore. The activated sample had a free pore volume of 2097 Å³/ unit cell. Upon loading with CO this reduced to 1782 Å³ / unit cell, and cooling reduced it slightly to 1770 Å³ / unit cell; this is approximately 50% of the total, theoretical, dehydrated pore size.”

Comment: This presumably refers to when the ligated water molecule is deleted. According to MERCURY (for which values will be slightly different) the pore volume is then 2353 Å³ / unit cell, thus the pore volume of CO loaded at 300 K is much more than 50% of the theoretical dehydrated pore size.

Regarding the pore volume percentage, the value calculated by crystal maker was compared with Olex and mercury and found to be too high due to an unrealistic estimation of what parts of the structure would be available. This has been corrected by using the lower value calculated by Olex. The values have been updated in the paper and supplementary information.

2. The authors stated:

“The e- density in the pore of the activated sample was estimated at 0.147 e-/ Å³. Loading with CO increased this to 0.535 e-/ Å³ and cooling slightly reduced it to 0.531 e-/ Å³. This is the expected pattern as introducing CO into the pores will increase the e- density, cooling increases the amount of CO chemisorbed so there is now less free in the pore. From these values it can be estimated that there are roughly 24 CO molecules per unit cell tumbling freely within the pore at 2 bar pressure and 300 K.”

Comment: I need some assistance in understanding the numbers. For CO loading at 300 K, the electron density was estimated to be 0.531 e-/ Å³. I presume this is multiplied by the quoted pore volume of 1770 Å³ in order to arrive at the total number of electrons? In other words, not the multiplied by the unit cell volume? If the former holds that will yield ~940 electrons for the pore per unit cell. CO has 14 electrons, how does this number correspond to 24 CO molecules per unit cell, because 14 x 24 = 336? The disparity is even worse if the residual electron density per Å³ is based on the unit cell volume and not the pore volume. In this regard, I suggest that the authors add a section in the supplementary information to show how they have arrived at the estimation of the number of physisorbed gas molecules.

Regarding the molecules per unit cell, there was an error in this calculation. We had used an incorrect value for the number of electrons within CO, this has been corrected within the paper. In addition, a small summary of how the calculation was performed has been added to the supplementary information.

We would like to thank reviewer 3 for his thoroughness in reviewing this work.